# Policymakers’ Perspectives Towards Developing a Guideline to Inform Policy on Fetal Alcohol Spectrum Disorder: A Qualitative Study

**DOI:** 10.3390/ijerph16060945

**Published:** 2019-03-15

**Authors:** Babatope O. Adebiyi, Ferdinand C. Mukumbang, Lizahn G. Cloete, Anna-Marie Beytell

**Affiliations:** 1School of Public Health, University of the Western Cape, Cape Town 8001, South Africa; mukumbang@gmail.com; 2Division of Occupational Therapy, University of Stellenbosch, Stellenbosch 7602, South Africa; lizahn@sun.ac.za; 3Department of Social Work, University of the Western Cape, Cape Town 8001, South Africa; ambeytell@uwc.ac.za

**Keywords:** policies, guidelines, fetal alcohol spectrum disorder, policymakers, interventions, services, women, developmental disabilities, alcohol, children

## Abstract

Fetal alcohol spectrum disorder (FASD) has a high prevalence in South Africa, especially among the poor socioeconomic communities. However, there is no specific policy to address FASD. Using a qualitative study design, we explored the perspectives of policymakers on guidelines/policies for FASD, current practices and interventions, and what practices and interventions could be included in a policy for FASD. The data analysis was done using the Framework Method. Applying a working analytical framework to the data, we found that there is no specific policy for FASD in South Africa, however, clauses of FASD policy exist in other policy documents. Preventive services for women and screening, identification, assessment, and support for children are some of the current practices. Nevertheless, a multi-sectoral collaboration and streamlined program for the prevention and management of FASD are aspects that should be included in the policy. While there are generic clauses in existing relevant policy documents, which could be attributed to the prevention and management of FASD, these clauses have not been effective in preventing and managing the disorder. Therefore, a specific policy to foster a holistic and coordinated approach to prevent and manage FASD needs to be developed.

## 1. Introduction

Alcohol is identified as the primary cause of preventable birth defects and developmental disorders [1]. Fetal alcohol spectrum disorder (FASD) is an umbrella term for the four categorical diagnoses in one diagnostic schema and a diagnostic entity for the adverse effects of prenatal alcohol exposure in another. The four categorical diagnoses of FASD include fetal alcohol syndrome (FAS), partial fetal alcohol syndrome (pFAS), alcohol-related neurodevelopmental disorder (ARND), and alcohol-related birth defects (ARBD) [2].

South Africa has the highest alcohol consumption rate (11 liters per capita) in Africa and among the highest in the world [3]. The lifetime consumption of alcohol for men and women is 49% and 22%, respectively [4]. The consumption rate of women seems low, yet those who consume alcohol do so in excess, and binge drinking is rampant [5,6,7]. Therefore, it is no surprise that in South Africa the national prevalence of FASD ranges from 29 to 290 per 1000 live births [8]. While the national prevalence is comparatively high, the prevalence of FASD in the Northern and Western Cape Provinces has received particular attention. In the Northern Cape, the prevalence of FASD was 88 per 1 000 in grade one learners in 2008 [9]. In 2015, the prevalence had dropped to 63.9 per 1000 in grade one learners, although still relatively high when compared to the national average [10].

In the Western Cape, the prevalence of FASD in grade one learners was estimated at 89.2 per 1000 in 2007 [11]. By 2013, the prevalence had doubled (135.1 to 207.5 per 1000) [12], whereas it had increased to 170 to 233 per 1000 in grade one learners in 2016 [13]. These estimates indicate a persistent rise in the prevalence of FASD in the Western Cape, indicating the need for context-specific preventive interventions and early intervention. Although these studies were conducted at different sites, these areas had similar characteristics (rural and lower socioeconomic status).

FASD is recognized as an important public health problem in South Africa [14]. Consequently, various researchers have advocated for services that target FASD in various forms. Some of these recommendations include context-specific prevention programs, the creation of adequately resourced mental health facilities for supporting the mental health needs of women before and during pregnancy and individuals diagnosed with FASD [15,16]. However, appropriate prevention and management efforts continue to pose a challenge for policymakers and service providers.

According to Pienaar and Savic [17], there is a paucity of local research for evidence-based interventions proposed by the National Drug Master Plan (2013–2017). Therefore, they suggested that lived experiences of alcohol and other drug (AOD) consumers should inform the development of policy. Interventions such as universal prevention, motivational interviewing and service provider short courses for FASD prevention which have been found effective should also be included [18,19,20]. However, these interventions have not been routine for FASD prevention or have low coverage because of the inadequate health system [21]. Jacobs et al. [14] acknowledged that efforts were being made to develop relevant policies to address the use of alcoholic beverages by pregnant women in South Africa, but the authors stressed the need to address the gaps related to the conspicuous absence of issues around FASD. Adnams [22] echoed the concerns of Jacobs et al. [14], advocating for partnerships and collaboration to address these gaps.

Rendall-Mkosi et al. [23] reviewed the South African National Government policy documents to explore the extent to which FASD is addressed. The authors found that only two documents referred to FAS and another one to women and alcohol consumption during pregnancy. Rendall-Mkosi et al. [23] also found that despite the absence of specific policies on FASD, the Departments of Health, Education, and Social Development were offering services to individuals with FASD. Although, these services are generic to people living with mental diseases and not specifically targeting individuals with FASD.

In light of the absence of a coordinated effort targeting FASD, we identified the need to develop a guideline to inform a multi-sectoral and interdepartmental policy to address FASD. In this article, we report on a step towards developing a guideline to inform a coordinated approach in the prevention and management of FASD [24]. We explored the perspectives of policymakers on existing guidelines/policies for FASD, current practices and interventions, and what practices and interventions that could be included in a policy for FASD.

## 2. Methodology

### 2.1. Study Setting

This study was conducted in the Western Cape Province of South Africa, which has the highest FASD prevalence in the world [13] although it is only the fourth largest and the third most populated province of South Africa. Within the Western Cape, FASD is most prevalent in the low-income areas, where binge drinking is rife, and the alcohol trade is largely unregulated. The Western Cape has a unique historical drinking culture entrenched in a system known as the ‘dop’ system—whereby the wages of farmworkers are paid using alcohol beverages [25]. Although the system has been abolished, the lingering effects remain. To this end, risk factors for alcohol and substance misuse increase, especially among populations with a low educational level and/or employed in menial jobs. Poor mental health and risky drinking practices, especially among women of childbearing age, are some of the many predisposing factors for FASD and these factors are endemic in this province [26].

The Western Cape has an estimated 6.5 million residents [27] with an infant mortality rate (IMR) of 19.1 per 1000 live births and an under-five mortality rate (U5MR) of 24.1 per 1000 live births [28]. According to Statistics South Africa [4], 38.1% women in the Western Cape reported having drunk alcohol once in their lifetime, while 27.3% had drunk alcohol in the preceding 12 months. Furthermore, 18% reported having drunk alcohol the preceding seven days, and 9.0% reported having drunk five or more drinks at least on one occasion the preceding 30 days.

According to a study conducted in the Cape Metropole, Western Cape, out of 684 pregnant women sampled, 36.9% confirmed that they had consumed alcohol during their current pregnancy or in the three months before they knew they were pregnant [29]. “In another study conducted in Cape Town with 110 samples (parents or caretakers of 110 children aged 4–12 years), the reported alcohol use six months before pregnancy and during pregnancy was 38% and 15%, respectively [30]. In proxy respondents, alcohol use six months before pregnancy and during pregnancy was 47% and 33%, respectively. The disparity in the percentages recorded is attributable to the stigma and shame associated with drinking during pregnancy [30]. In the Western Cape, women also considered drinking alcohol during pregnancy as a coping strategy for their sociopolitical realities [31].

In the Western Cape, liquor-related commercial activities are the second most frequent category of business in the township economy [32]. These liquor-related businesses are identified in the form of shebeens/taverns, and an estimated 25,000 of unlicensed/illegal liquor stores are operating in the Western Cape. About 60% of these shebeens/taverns are run by women selling low volumes of alcohol to complement their household income. This trade has made alcohol available to people of all ages.

### 2.2. Study Design

We employed an exploratory qualitative design. Information gathered from qualitative research adds value to the development and implementation of guidelines [33] as it allows for an in-depth exploration of the perspectives of relevant stakeholders [34].

### 2.3. Sampling Procedure

A two-step purposive sampling approach was adopted. Step one involved selecting three institutions and step two involved selecting 10 participants who were policymakers on FASD-related service delivery. In step one, the Western Cape Department of Education, and Departments of Health and Social Development were selected as the most relevant stakeholders (policymakers) for the development of policies, interventions, and services for individuals with FASD. In step two, we adopted the following criteria to select the study participants, (1) participants working in any of the above-mentioned Departments; (2) and having at least five years’ experience in policymaking or services and implementing interventions for individuals with FASD; (3) or were members of the multidisciplinary team working on FASD. The characteristics of the study participants recruited for the interviews are described in Table 1.

Potential participants were directors, assistant directors, heads of programs, policy administrators, policy developers, and policy monitoring and evaluation officers. All the participants had tertiary qualifications as the lowest qualification. They were recruited via emails, telephone calls, and personal visits. Fifteen potential participants were identified. Twelve agreed to participate in the study, but only 10 were finally interviewed. While conducting the interviews, we reached data saturation with the subsequent responses and discussions, as no ‘new’ information was elicited [35,36].

### 2.4. Data Collection

In-depth interviews lasting 30–60 min were conducted in English between September 2016 and 2017, and guided by an interview schedule (Appendix A). We used open-ended questions to start the interviews and follow-up questions to probe for additional explanations when required. We asked the study participants various questions on available policies on FASD, the coordination of FASD interventions, and relevant aspects of FASD to explore information that could inform the development of appropriate guidelines. All the interviews were audio-recorded with permission from the study participants.

### 2.5. Data Analyses

The data were analyzed using the Framework Method [37], a part of the thematic analysis family [38]. Following the Framework Method, we transcribed the interviews, and read and re-read for familiarization. The transcripts were then coded inductively to generate initial codes and re-organized to obtain refined codes. Using the initial codes, two of the authors (B.O.A. and F.C.M.) and an independent coder developed a working analytic framework (Figure 1). Codes with similar concepts were grouped to form sub-themes, and those sub-themes, which addressed similar concepts, were further grouped to form the final themes.

The working analytic framework we developed was applied by attributing relevant aspects of the texts in the subsequent transcripts to existing categories, themes, and sub-themes. Finally, we charted the data into the framework matrix (Table 2 and Table 3).

### 2.6. Trustworthiness and Rigor of the Study

The rigor and trustworthiness were established through credibility, transferability, dependability, conformability, and reflexivity [39].

During data collection, we guarded against leading the respondents and allowed them to express themselves in sharing information. We provided a detailed methodology for the study. Verbatim transcripts of the participants’ responses were included to ensure credibility. Furthermore, credibility was enforced through member-checking at the end of each interview—a recap of the salient points that emanated from the interviews. In addition, a reflective journal, which documented the discussions, deliberations, and decisions of the researchers was kept during the entire study. The reflective journal formed part of the audit trail that was kept for the study.

In consultation with the research team and based on the literature on FASD, the interview schedule was developed in a way that allowed the required information to be obtained to ensure dependability. The same interview schedule was used as a guide for all the interviews. Two members of the research team independently coded the transcripts and met afterward to discuss the findings. An agreement was reached by consensus.

The research team (a PhD student, an expert on FASD and two supervisors) has no personal relationship with the participants, however, the university that approved the study has a working relationship with the government departments.

In reporting this study, all the relevant aspects of the criteria for reporting qualitative research (COREQ) outlined by Tong, Sainsbury, and Craig [40] were followed.

### 2.7. Ethical Considerations

The approval for the study was obtained from the research ethics committee of the University of the Western Cape (BM/16/4/4), and further approvals were obtained from the Western Cape Department of Education (20161212-6937), Department of Health (WC_2016RP29_862), and Social Development (12/1/2/4).

Before conducting the interviews, the study aims and objectives were explained to the potential participants who were provided with an information sheet written in English explaining their roles. The potential participants were requested to sign a consent form as they agreed to participate in the study. Because some of the participants are in prominent administrative positions within their departments, there is a chance that these individuals could be linked to their responses. The researchers warned all the study participants of this potential before they signed the consent forms. All information obtained during the study was kept strictly confidential on a computer with a password known only to the research team.

### 2.8. Data Management

To maintain anonymity and easy identification of the information sources, policymakers in the various departments are coded as DOE_X_, DOH_X_, and DSD_X_. X denotes an arbitrary number from 1–4.

DOE represents Department of Education, DOH denotes Department of Health, and DSD denotes Departments of Social Development.

## 3. Results

Presented below are the salient categories, themes, and sub-themes obtained during the data analysis.


**Category 1: Availability (Lack) of guidelines/policies on FASD**


All participants from the three selected departments acknowledged the absence of policies/guidelines specifically addressing the issues around FASD.


*We [(at DOE) do not have any direct policy for FASD but the policy documents I gave you are all part of what we call inclusive education.*
(**DOE_2_**)


*We do not have a guideline per se for FASD. We (at DSD) fund non-profit organizations (NPOs) to render services. In our (DSD) annual performance plans, we (at DSD) have indicators as to what we need to achieve as a department.*
(**DSD_1_**)


*We (at DOH) do not have any guideline or policy specifically for FASD as such in our Department, so there is nothing to guide any intervention or screening anything specifically.*
(**DOH_2_**)

One of the participants suggested that the absence of a policy on FASD could be related to the fact that other ‘bigger’ issues such as mental health have not yet received the due attention to warrant a separate policy as a coordinated effort to address it.


*We [at DOH] do not even have a specific policy for mental health conditions but now it has been combined with clinical conditions, e.g., HIV, AIDS, and mental health are included, that it is [the only] health policy we have for now. We do not have any policy, but we did develop a specific policy last year, but it does not look at FASD condition. It is just into mental illness not into FASD…*
(**DOH_2_**)

Some participants, nevertheless, indicated that various clauses exist in different policies related to the prevention and management of FASD. The following participants identified aspects of the prevention and management of FASD that were found embedded in policies addressing other issues.


*We (at DOE) have a policy document on special education needs in terms of education. It is a very broad policy document but Fetal Alcohol Spectrum Disorder is one of the areas that are titled in the policy document… we have the barriers to learning (aspects of the) policies but that includes specific learning barriers, physical barriers (blind, deaf cerebral palsy children). Therefore, FASD is one of the categories …*
(**DOE_2_**)


*I am in women’s health and we (at DOH in our division) deal with maternal health. In maternal health, as part of antenatal care, we follow the National Maternity Care guideline and it speaks to the first antenatal booking. There is history taking that we must do and the part that history taking speaks to alcohol use as well.*
(**DOH_3_**)


*The FASD is part of a policy (alcohol-related harms reduction policy) but in a certain chapter like health and social services, education awareness and community action group. Therefore, it [FASD policy] is worked into our alcohol-related harms reduction policy, so it (alcohol-related harms reduction policy) is becoming a white paper. It (alcohol-related harm reduction policy) has gone through to the Cabinet and after Cabinet, it (alcohol-related harms reduction policy) could become a white paper—official and then it (white paper) would become policy.*
(**DSD_3_**)


**Category 2: Development of guideline/policy document**


The participants had different opinions on whether a separate guideline/policy should be developed for FASD. Those who opted for a separate policy argued that FASD is a serious problem, especially in the wine producing communities of the Western Cape Province.


*I would really advocate for a separate policy, separate attention given to FASD due to the severity of the problem. Because it is a severe problem in the Western Cape and if you look at communities, people have a greater need for a specialized service when it comes to FASD*
(**DSD_2_**)


*Guidelines will definitely be helpful because it will guide appropriate intervention from an early identification point of view. Therefore, if it is identified and properly managed through guidelines, you would probably end up having a more positive result in terms of returning the child to school and not ending up being a dropout.*
(**DOE_2_**)

Opponents for having a separate policy on FASD suggested that the implementation of such policies would be resource-intensive, which is impracticable in the context of scarce resources. In addition, these individuals argue that government departments are moving towards policies that are inclusive in nature, rather than having specified policies for various conditions.


*I believe everyone or anyone would like a specific policy to be developed for a specific issue. However, the first thing that we look at is if it is practical. Can it be implemented? Can the policy address the issue? In addition, can it serve the purpose that it is being developed for? However, for FASD, as I told you, in the beginning, we have schools for the blind, deaf children, schools for cerebral palsy children; we have schools for learners with slow cognitive thinking. We do not have special schools for FASD, but they will be catered for in the special needs schools. You see, as I told you, if there are funding and resources to develop a specific policy that will address this in education, and then I would advocate it. Then, I would say this could smartly be addressed in the department of health or social development, we (at DOE) are just trying to develop them [individuals affected by prenatal alcohol exposure] for the future.*
(**DOE_4_**)


*I am not sure if that (separate policy for FASD) is necessary either because in terms of the inclusive policies that child should be identified as having a barrier to learning. Therefore, if you want a policy on that it will cost the management because the teacher needs to know how to address children with those barriers and that may even be university level. I think maybe there is a need to have a module on how to deal with FASD children because they are going (to) be in the classroom.*
(**DOE_2_**)


**Category 3: Current practices and available FASD interventions**


This category presents current practices and interventions that have been pinpointed for the prevention and management of FASD. The participants identified several prevention programs that are currently run by various departments in the Western Cape.

Regarding FASD, a participant from DSD reported that there are ongoing awareness and prevention programs on alcohol and drug abuse under the special program of the Department of Social Development.


*So, it is basically just general awareness and prevention services when it comes to FASD. It’s an alcohol and drug abuse awareness program. And alcohol and substance abuse or drug abuse has been identified as a special program under Social Development.*
(**DSD_2_**)

Another participant said that in its annual performance plan, DSD has strategies to help mothers with alcohol problems. The participant reported that helping mothers with alcohol problems aims at preventing FASD, which is usually achieved by funding (NPOs).


*We (at DSD) have indicators in terms of what we need to achieve with regard to helping and assisting a patient (with an alcohol problem) or a client with FASD. (We assist the) mother first. We as a department (Social Development) fund organizations to do this.*
(**DSD_3_**)

The participant from the DSD also reported that there is an ongoing awareness program in various government departments such as the Western Cape Liquor Authority, and the Western Cape Departments of Agriculture, Education, Health, and Social Development.


*There is a lot of awareness going on at the clinical level as well as when someone comes to our (Western Cape) local clinics. They (at Western Cape local clinics) do early intervention and try to motivate the client (women) to seek further treatment for the alcohol problem. And then [the Department of Education is running] educational and awareness (programs) at schools, also, the Western Cape Liquor Authority is going to the schools to have road shows and showing school children the pro(s) and cons of drinking [alcohol].*
(**DSD_3_**)

A participant from the DOH indicated that there is a guideline at clinic-level for screening women for any drug use including alcohol.


*We (at DOH) have a PACK guideline (practical approach care kit), so this is just to help the clinician in the clinic. So, if you look at the women’s health, you screen for any drug [alcohol inclusive] abuse.*
(**DOH_1_**)

Various FASD-related management interventions that are currently applied in Western Cape departments were also identified by the participants.

A participant from the DSD indicated that services are rendered to individuals with FASD which are included in general social work services at the community level.


*FASD is more addressed on a local level, at a service delivery level. The actual services that are rendered to the people affected by FASD are included in the generic social work services at grass-root levels in the local offices of the Social Development.*
(**DSD_2_**)

The participants from the DOE indicated that their department provides educational training and support on FASD for teachers and the support team—psychologists, social workers, and support teachers. The DOE also provides training on how to identify and manage children with FASD in class.


*What we (at DOE) have is a commitment to training teachers in the area of the fetal alcohol spectrum disorder. What are the causes? What are their characteristics? In addition, what are the consequences for teaching and learning? Therefore, we have the training sessions across the province for fetal alcohol spectrum disorder.*
(**DOE_1_**)


*We (at DOE) are planning a training workshop for our psychologist and I am busy planning for the training with the NPOs, to train our psychologist on early identification and treatment of FASD children. We have asked that the training must [place] emphasis on early identifications. Some have to be according to DSM five (Diagnostic and Statistical Manual of Mental Disorders) because they must diagnose correctly. In addition, the emphasis must also be on how to advise the class teacher on how to teach the child in the classroom.*
(**DOE_2_**)

The following quotes illustrate the availability of referral pathways for the management of a child in schools: one within the Education Department and the other between the DOE and other government departments.


*If a child stays out of school for 10 consecutive days without any reason or explanation, we (at DOE) will call for deregistration and the child is taken out of school. However, the child is not lost; the child will be taken to social development to check if the child does not suffer from whatever special needs. After that, the Department of Social Development will be the parent or guardian for the child.*
(**DOE_4_**)


*In schools, teachers are involved in governance. However, when there is a problem and the teacher cannot solve it, the learning support person in the schools or district will help. In addition, if there is a need for (a) psychologist or social worker they will also help. We (at DOE) have outreach teams. They include social workers, psychologist(s), and learning support people; they help an ordinary schoolteacher. Moreover, we have a school-based support team that diagnoses problems in the schools.*
(**DOE_1_**)

Another participant from the DSD identified an alternative referral pathway through NPOs funded by the DSD. The participant indicated that the NPOs have a role in identifying the child and channeling him/her to the right school.


*The child obviously needs to be channeled into the right schools, be identified at home as well at the school… We as a department (of Social Development) fund organizations to do this.*
(**DSD_3_**)

A participant from the DOE indicated that it is the DOE’s responsibility to involve parents of a child with special needs, as it is part of the policy (Schools Act).


*So, parents’ involvement is crucial; if a child transgresses, his parents are always informed. It is part of the policy (School Acts)*
(**DOE_4_**)

Some participants from the DOE also reported that the Department of Education is currently busy with the screening, early identification, assessment, and support. The Department aims to identify children from the lowest grade possible—such as the reception grade—where they are received into the formal schooling system (grade R).


*We as a department (Education) are busy rolling out a process for the identification of the child from grade R level. And it is in two, firstly, there is a learner’s profile, which is the biographic details of the learners; some of these details may probably be an indication of FASD. Secondly, there are forms within which the teacher fills in a special needs assessment; form one, two and three.*
(**DOE_2_**)


*So, the screening, identification, assessment, and support (SIAS) program is to try to bring the identification down, the education spectrum possible. So, if we (at DOE) can try and pick this up as early as grade R.*
(**DOE_3_**)

A participant from the DOH reported general service for the management of FASD.


*Services (for individuals with FASD) are integrated. It (service for individuals with FASD) is dealt with like any other child requiring a service or mother who has a substance problem and requires a specialized health service.*
(**DOH_1_**)

In Table 2, the different practices, interventions, and programs that are currently in place are illustrated according to the participants for the prevention and management of FASD.


**Category 4: Identified policy requirements for FASD**


This category presents the suggestions for FASD policy.

The participants suggested that an FASD-related guideline should facilitate early identification, screening, and diagnosis. They advocated that such a guideline would facilitate proper diagnosis, early intervention, and tailored response. The guideline would also ensure the development of diagnostic services with proper measures for accurate diagnoses by qualified health professionals.


*Earlier identification and screening for everybody; not just FASD children, all disabilities. What I am saying if the Department of Health should pick it up in the early years when they are doing the developmental screening and if there could improve communication between the Department(s) of Health and Education. We will address the FASDs issues better.*
(**DOE_2_**)


*Yes, early identification… not to put the diagnosis in the hands of the teachers…. I think our biggest problem is that they are not diagnosed. So, the child may be referred to us for concentration problems but actually it’s [an] FASD problem. This is because they are referred to us for behavior (-related issues) but actually, they have a neurological disorder, which we do not know about yet.*
(**DOE_3_**)


*What we need first, is the tool to screen and identify, if we are to identify and collect data. Then, from there we can use this evidence to put measures in place…*
(**DOH_1_**)

Another participant from the DSD suggested that the needs of individuals with FASD should be included in all social welfare plans, and the functioning of those with FASD should be looked at within the family system.


*So, the whole social service sector should really realize that FASD is the problem and that it should be included in all our [social development] social welfare plan(s)… We should look at the whole functioning of the person with FASD in the family [as] very important. So, then we should look at the social, emotional, psychological, educational and inspirational aspects of the person’s life and obviously the person with the FASD within his family environment.*
(**DSD_3_**)

A participant from the DOH suggested that assistance should be given to women or pregnant women and a referral pathway for help should be made available.


*Firstly, in pregnancy, they (women) need assistance with a safe house and referral pathway. (For example), if I am a heavy drinker and I realize today that I am pregnant, but I feel the responsibility to have my child protected, what is in place for me now to stop drinking? If I am living in an abusive relationship, where I am drinking with my husband? What is in place for me if I want to stop drinking today and leave this abusive drinking husband? So, that I can safeguard my child. Also, the referral pathways.*
(**DOH_1_**)

Another participant suggested that the pathway for managing a child with FASD should be included in an FASD-related guideline.


*If a child is born and it is determined by the Health Department and Social Development that the child has specific disorders, then the child is going to be referred to a specialized school. This (is) done towards educating the child as far as possible. So that the child can fit into the community or into the work environment if it is possible.*
(**DOE_4_**)

A participant from the DOH suggested the need for early intervention and the need to adopt relevant aspects from other policies like Western Cape Government First 1000 days.


*We need remedial interventions early on. We focus a lot on the first thousand days of the brain development, eating and overall care of the child and if the identifying measures can start early on. This might also assist…screening pregnant women and helping other people with intervention.*
(**DOH_1_**)

The participants suggested that a policy guideline could facilitate collaborations among relevant departments and sectors like the Departments of Community Safety, Health, Education, and Rural and Land Reform, the Western Cape Liquor Authority, the South African Police Services (SAPS), the City of Cape Town and the Municipalities. The guideline should also elucidate the roles and responsibilities of the relevant stakeholders with specified and proper channels for collaborations established. The participants advocated that the guideline should illustrate how FASD services should streamline and connect through the developmental stages of an affected child.


*Guidelines will definitely be helpful because it will guide appropriate intervention from an early identification point of view. So, if it is identified and properly managed through guidelines, you would probably end up having a more positive result in terms of returning the child to school and not ending up being a dropout. You may say at present, each department works on its own… when you speak about Education, the DSD, and the Health; you need to look at a new guideline [that will make us work together]. (A guideline) that will look at how these three departments will connect in transferring information from one department to another department. This is because it is like, we (at the Department of Education) get the child, and we enroll the child. The child starts in grade R and we have no medical history (of the child). We need to track the multi-development (of the child), we need to track the physical (mental, social and others) development of the child, but now we (at the Department of Education) do not have that.*
(**DOE_2_**)


*I think that [inter-departmental collaboration] is definitively the way to go because there are many issues that each Department is dealing with in their own way. This is because if they (departments) can collaborate more and have a relationship with each other, then, it will be better. So, definitely, there is surely a need for multi-sectoral collaboration. Maybe for Social Development, for example, to look at the social need of a child, (Department of) Education looks at educational need and so on. So, to have a conversation between all the departments definitely on how services can be streamlined, it will make it easier to connect them all.*
(**DOH_2_**)


*It will not be of any help if the Department of Social Development develops an FASD policy that only applies to our department. So, it (the guideline/policy) needs to cut across all departments and departments follow the same policy or at least we (Departments) fit in into it [the guideline/policy).*
(**DSD_2_**)

Some participants from the DSD expressed the need for all the Departments and other role players to be involved in addressing the FASD issues. This is because each of them provides a specific service or services that are beneficial to individuals with FASD. The participants echoed that the guideline should be designed in such a way that will bring these services together.


*Basically, all departments like Department of Community Safety, Department of Health, Department of Education, Department of Rural and Land Reform, the Western Cape Local Authority, the South African Police Services and other key role players like your Municipalities; for example, Cape Town should be involved. For instance, all of them are involved in the alcohol harms reduction policy.*
(**DSD_2_**)


*The policy care for people with disabilities…. you will find child care and protection, you will find your FASD empowerment program, you will find current prevention program, you will find youth program. All I am saying is that irrespective of the social program, it must remain accessible to all people including the persons who have disabilities… we (Social Development) will be guided by the integrated service delivery model. Our service delivery, integrated service delivery is divided into four levels. One is awareness, the second is early intervention, the third is statutory and the fourth is reunification.*
(**DSD_1_**)

Some participants from the DOE suggested the need for FASD-related training for parents and teachers.


*There is a huge problem related to parental training. It is a massive problem because you would have to train every parent and I am thinking in a lot of cases—individuals with FASDs are coming from lots of areas.*
(**DOE_2_**)


*I think we need to advocate for parental training, early diagnosis, and preschool identification. If we are looking at school policy again, early identification allows us to put them into the program early; and teacher training, train teachers to deal with these children because they are in the classroom.*
(**DOE_4_**)

Participants also suggested the provision of inclusive education and that a separate school for children with FASD may not be necessary as this will be contrary to inclusive education. They advocated for more specialized services and support for children with FASD within the mainstream school.


*I think it would be contrary to the inclusive education (to have FASD children in special schools). This is because the inclusive education says that we should try to assist the child in the area in which they live. So, we do not want to unnecessarily remove young children from home and place them in a boarding home. So, the inclusive policy, when you read the full services, [the child should be put in] the normal mainstream school in the community, which has additional support.*
(**DOE_2_**)


*So, I think the contact of the child for me is so crucial because often we think of just removing the child from one school, placing the child at another but we never discuss it with the child…removing a child from one school to another is traumatic. This is because the child loses his friends, he loses his teacher—a teacher comes with teaching styles, with different teaching styles—he loses his peer, he loses his safe space and then you put the child into a different school. I mean we need to accommodate the interest of the child and we need to communicate and discuss it with the child.*
(**DOE_4_**)

All the various practices, interventions and programs that the study participants identified which should possibly be included in policy for the prevention and management of FASD are indicated in Table 3.

## 4. Discussion

We aimed to identify relevant FASD preventive and management practices and interventions based on the perspectives of policymakers to inform the designing of a guideline for FASD policy. Our findings indicate that there are no specific or separate national policies/guidelines for the prevention and management of FASD in South Africa. However, there are documents on policies/guidelines indirectly addressing the issue of FASD in a generic way [23]. For instance, the National Drug Master Plan, the National Human Genetics Policy Guidelines for the Management and Prevention of Genetic Disorders, Birth Defects, and Disabilities and the Guidelines for Maternity Care in South Africa all have prevention and management modalities that could be identified as related to FASD. In addition, the Education White Paper 5 on Early Childhood Development, Education White Paper 6 on Inclusive Education, South African Schools Act, Western Cape Provincial Schools Act, and the recently drafted Western Cape alcohol-related harms reduction policy also contain elements of FASD prevention and management.

The policies mentioned have not been systematically evaluated to assess the extent to which they address issues regarding FASD [23]. The National Drug Master Plan (2013–2017) proposed the use of evidence-based interventions, however, there is a paucity of local research that could inform possible interventions [17]. Furthermore, existing interventions that are specific to FASD are not addressed by the above policies. Specific interventions in the different sectors have not been made routine for FASD prevention [18,19,20], and sometimes the financial aid for these interventions is not sufficient [21].

Moreover, although the mentioned policies/guidelines generically cover aspects relevant to FASD, there is no specific national document (policy or guideline) that exclusively and comprehensively focuses on FASD. The lack of FASD policies has a negative influence on the delivery of FASD specialized services. The absence of a specific or separate policy/guideline for the prevention and management of FASD has also been reported by Rendall-Mkosi et al. [23]. In our study, we found that developing a specific FASD policy that requires huge capital and an additional workforce could be one of the reasons for not having a specific policy. Another reason could be the inadequate and overburdened health system in South Africa [21].

Although it has been reported that FASD is a serious public health issue, particularly in the Western Cape Province and the wine producing areas [8,41], we found divergent opinions on whether or not a separate FASD guideline/policy should be developed. Most of the policymakers agreed to the need for a specific policy for FASD, while a few did not see the need to develop guidelines pertaining exclusively to FASD-related issues. They cited high cost as a barrier to developing and implementing such policies. In one accord with these participants, other authors have expatiated that having a separate policy required additional workforce and implementation cost, thereby increasing the burden on an already overwhelmed health, social and educational system [42,43,44,45]. The notions expressed by the policymakers could indicate that the relevant departments should work together collaboratively to do more, which may not necessarily translate to increased costs for developing and implementing an FASD policy [46,47].

We found that there are general awareness programs/services at the local level (community-oriented programs). These awareness programs are carried out as part of the general social work services rendered to the people in the community. In general, these awareness programs focus on drug and substance use/abuse suggesting that preventive efforts for FASD may not necessarily require specialized programs [48,49]. However, there are few prevention programs targeting women and pregnant women [15,16] at clinics with the aim of preventing FASD. We discovered that the DSD usually funds NPOs to provide prevention programs. There are also educational and awareness programs such as roadshows at schools. These are being carried out by the DOE and Western Cape Liquor Authority.

Although there are several current preventive interventions/practices reported, these have not been effective in reducing the prevalence of FASD in South Africa, especially in the Western Cape [50]. The ineffectiveness of these interventions could be attributed to other mitigating factors such as poor living conditions, poor nutrition, poor socioeconomic status, low-level of education, low-income, and unemployment [13]. Therefore, there is a need for comprehensive preventive services addressing gaps in current policies and policy implementation to facilitate strategic interventions [14,22].

Regarding the management of FASD, the findings indicated that services are also rendered to individuals with FASD as well as their families, as part of generic social work services at the local level. The DSD sponsors organizations to render services to individuals with FASD and their families. However, the problem with this is that such services only address the social issues neglecting the medical and educational issues, which are equally important for the holistic management of FASD. Therefore, there is a need to develop a national policy that will facilitate the coordinated management of FASD across the relevant departments.

We found that in the DOH, based on available policies, services such as general developmental screening, diagnosis, developmental stimulation, and general medical management are part of integrated services, and there are no specialized services for individuals with FASD and their families. While the integrated services could place less burden on the health system, the effectiveness of these services may be undermined if the social and educational issues are not addressed concurrently.

We discovered that in the DOE, FASD is considered as one of several causes of learning disabilities. Thus, the DOE has policies addressing learning disabilities rather than focusing on FASD. The problem with defining FASD based on one’s ability to learn is that individuals with FASD do not only have learning disabilities, these learning disabilities could lead to other pertinent concerns regarding their well-being becoming neglected. Therefore, policies focused on addressing the educational needs of individuals will inherently fail to address the broader social issues of individuals with FASD [51].

In addition, one of the policies relates to screening, identifying, assessing, and supporting people with learning disabilities from grade R [52,53,54]. These individuals with learning disabilities are supported from when they are very young, as early intervention has shown promising outcomes [55]. The DOE also provides training on FASD and other learning disabilities (identification, consequences for teaching and learning, and classroom management) for teachers, social workers, psychologists and members of the support team.

In our study, we showed that efforts are being made to use a multi-professional support team for children with special educational needs (including children with FASD) starting from school to the provincial office. When a child is suspected of having FASD-related learning disabilities or any other learning difficulties, he/she is referred to a school-based support team. If the school-based support team is not equipped to resolve the issues, the child will be referred to the district-based support team, and may finally be referred to the provincial-based support team. They will assess the child and determine if he/she should continue in the mainstream or be moved to a special school. The inadequacy of these efforts is that the special education needs of these children are enormous; there are inadequate resources to meet these needs and those of their teachers and the support team [56]. Because of this, teachers may not be adequately trained to handle the emotional, social and other issues faced by individuals with FASD. The DOE also involves parents in the management of individuals with FASD.

Apart from the current practices and programs, we also identified other important aspects regarding the prevention and management of FASD that could be included in a guideline/policy in our study. The DOH and DSD should provide preventive services to women, pregnant women, and the community [23]. A child should be screened and diagnosed at an early stage by the DOH; this should be communicated to the DOE towards educating the child as far as possible. The DOE should communicate to the Department of Labor towards providing appropriate employment or augmenting the child’s education with necessary skills concerning employment or entrepreneurship. Consultations with the DSD should be done to assess the suitability of the child for social grants or any other social welfare packages. The guideline/policy should also facilitate the provision of inclusive education in line with the inclusive education policy [23].

Some of the suggestions made on what should be included in the policy were aligned with the four-part framework by pan-Canadian prevention experts proposed by Poole et al. [57]. This four-part framework consists of four levels of prevention for FASD. Level 1 involves comprehensive awareness building and health promotion efforts. Level 2 relates to discussions on alcohol use and related risks with all women of childbearing years and their support networks. Level 3 suggests specialized, holistic support of pregnant women with alcohol and other health/social problems. Finally, level 4 encompasses postpartum support for new mothers and support for child assessment and development.

We discovered that some of the current practices and available interventions suggested by the policymakers were similar to those identified by FASD service providers in another study [58]. These interventions and policy requirements included training and support for parents, caregivers, and teachers, multi-sectoral/inter-departmental collaboration, and general awareness in the community and schools. Not only that, screening, early identification and diagnosis and providing assistance for women with alcohol use problems were also part of the interventions and policy requirements mentioned. These interventions and policy requirements were also aligned with policy developed to tackle FASD in Canada and Australia. These policies include the Canadian Framework for Action on FASD and Australian National FASD Strategy for Action Plan 2018–2028 [59,60].

Most of the approaches identified in our study are classified as downstream and midstream approaches to addressing FASD. Arguments have been made that addressing FASD-related issues at the level of social determinants of health can improve health and reduce disparity [61]. Consequently, calls have been made for the use of an upstream approach (referred to as the second approach) for the prevention of FASD [62]. Therefore, for an FASD policy to be comprehensive and holistic, upstream prevention approaches should also be considered.

### Strengths and Limitations of the Study

In this study, the researcher interviewed policymakers from the three major departments (Education, Health, and Social Development) that are responsible for the implementation of policies and the provision of services to individuals with FASD. A limitation of the study that was expected to inform policy on FASD nationally was that it was only conducted in one of nine provinces in South Africa. We also did not interview policymakers from the Department of Labor and the Department of Justice, who may have had additional information to improve the guideline. Furthermore, we used purposeful sampling in the study, which may have prompted biased opinions. Moreover, since the focus of this study was on policymakers, we did not include women who drank during pregnancy, women with children who have FASD, women who went through pregnancy and did not drink, individuals with FASD or women with prenatal alcohol use who quit drinking upon hearing they were pregnant.

## 5. Conclusions

There is a consensus among relevant policymakers that there is no specific guideline/policy document on FASD. Many of the participants agreed that a national guideline/policy should also be developed for the holistic prevention and management of FASD. However, some of the participants suggested that developing a national policy that is exclusive to FASD and comprehensive to address all relevant aspects will be costly. In the study, we identified some aspects for the prevention and management of FASD that are being applied at various health system levels including those that could be added to build a guideline to inform policy about FASD.

## Figures and Tables

**Figure 1 ijerph-16-00945-f001:**
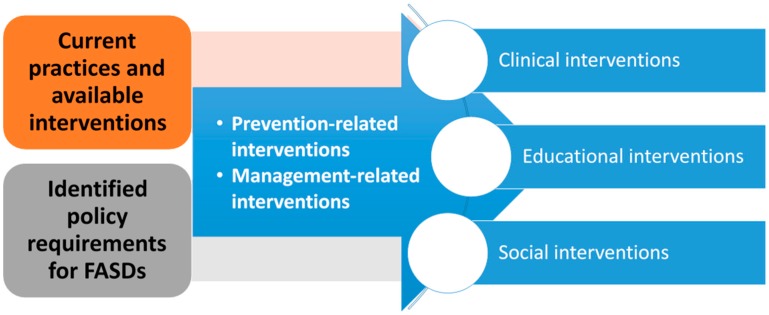
A working analytic framework.

**Table 1 ijerph-16-00945-t001:** Characteristics of study participants.

Characteristics	Participants (*N* = 10)
Number of interviews per department	
Department of Education	4
Department of Health	3
Department of Social Development	3
Gender	
Male	4
Female	6
Profession	
Allied health	5
Others	5
Working Experience (Years)	
5–10	3
11–20	3
21–30	2
31–40	1
41–50	1

**Table 2 ijerph-16-00945-t002:** Current practices and interventions.

**Prevention practices and interventions**	**Clinical**	Use of non-profit organizations [NPOs] to assist pregnant women with a drinking problem
		Referring women with an alcohol problem to DSD
		Taking alcohol history during antenatal booking
		Health promotion on alcohols for women in clinics
		Customized message on the risk of alcohol consumption
		Multi-sectoral collaboration for alcohol abuse prevention
		Motivational counseling to stop drinking
		Alcohol screening for high-risk women
	**Educational**	Prevention awareness in schools on alcohol
	**Social**	Funding NPOs to carry out FASD prevention program
		Awareness and prevention of alcohol abuse at a local level
		Motivating women to participate in an alcohol program
		Awareness schools and clinics on the danger of drinking alcohol
		Alcohol/drug abuse awareness program in the community
**Management practices and interventions**	**Clinical**	General developmental screening for children
		Use of NPOs to carry out a diagnosis for children
		Multi-sectoral collaboration for management
	**Educational**	Educating children with FASD and preparing them for the future
		Parent involvement in the management of a child with FASD
		Commitment to training teachers and members of the support team on the identification and management of a child with FASD
		Availability of outreach team to support teachers and learners
		Adaptation of curriculum for an individual with FASD
		Screening, identification, assessment, and support for children
		Provision of specialized education for individuals with FASD
		Specialized support for individuals with FASD within the mainstream school
		Availability of policy that addressed curriculum, behavior, and governance
	**Social**	Early intervention and aftercare assistance for children
		Availability of protective workshops for adults with a disability
		Availability of daycare service for children with disability
		Availability of residential services for children with disability
		Funding NPOs to provide residential care for children with FASD

**Table 3 ijerph-16-00945-t003:** Identified policy requirements.

**Prevention policy requirements**	**Clinical**	Need for multi-sectoral collaboration for prevention
		Facilitate local initiates to solve alcohol problems
		Establishment of a working group for policy development
		Emphasis on no alcohol is safe during pregnancy
		Need to screen women for alcohol and drug abuse
		Need for a prevention program for the women, the family, and the community
	**Educational**	Put a law in place to deter people from abusing alcohol
		Awareness about the danger of drinking alcohol
		Targeted intervention in high incidence area
	**Social**	Advocating for the right of individuals with FASD
		The inclusion of the needs of individuals with FASD in the social welfare plan
		Skill building for individuals with FASD
		Alcohol pricing and tavern regulation
		Create a local alcohol action committee
		Consultation with relevant implementation stakeholders
**Management policy requirements**	**Clinical**	Identification of the needs of individuals with FASD
		Need for a safe house for pregnant women and referral pathways
		Need for seamless connection with services on FASD problems
		Need for helpline that people can call for information on FASD
		Need for multi-sectoral collaboration for managing FASD
	**Educational**	Interdepartmental involvement (South African Police Services [SAPS], churches, and non-profit organizations [NPOs])
		Need for specialist care for individuals with FASD
		Need for a structured and dedicated program for FASD
		Training of teachers, social workers, and psychologist on FASD
		Need for inclusive education for individuals with FASD
		Classroom management guideline for FASD
		Pharmacological management of FASD
		Parents should create a care mechanism for their children.
		Early diagnosis and preschool identification for children
		Need for early intervention for an individual with FASD
		Need for school curriculum adaptation and functional curriculum for an individual with FASD
		Need for additional support within the mainstream school FASD children
		Need for medical assistant in schools for FASD children
		Early identification and prevention
		Clear roles and responsibilities of the different departments of management of FASD
		Appropriate child placement for FASD children
		Clear channels of communication among departments
		Proper communication to the parent about their children
		Communication with FASD children
	**Social**	Need for specified and appropriate intervention for different ages
		Training of caregivers, social service practitioners, and volunteers
		Promoting independent living
		Build residential facilities and daycare
		Care for individuals with FASD and their families
		Funding NPOs to provide personalized services for FASD

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
