# Peer review of "Policymakers’ Perspectives Towards Developing a Guideline to Inform Policy on Fetal Alcohol Spectrum Disorder: A Qualitative Study"

_ijerph, 2019, doi:10.3390/ijerph16060945_

Round 1

Reviewer 1 Report

Summary: The current manuscript by Adebiyi et al., aimed to investigate professionals’ impressions regarding policies on Fetal Alcohol Spectrum Disorders in South Africa. Overall, the authors found that there is concern and a clear need for new policies at various levels. The manuscript is a bit long because samples of many of the responses are provided. However, it is generally written well and is easy to follow. Below are a couple of suggestions to improve the quality and impact of this manuscript:

1.     In line 69, what does “women in relation alcohol consumption” mean? Perhaps there is a missing word.

2.     In line 91, “and this factors” needs to be changed to “and these factors”

3.     It would useful to include the level education from the participants, such as highest degree earned, in addition to "profession" that is already provided, in the methods section

4.     In line 169, “transcriptions” needs to be changed to “transcripts”

5.     In line 241, the word “naysayers” is a strong word to describe those who feel that it would be resource intensive to put new policies in play. Perhaps using "individuals" would be more professional and less judgmental.

6.     In line 261, change “… regarding FASD, there are an ongoing awareness …” to “… regarding FASD that there is an ongoing …”

7.     In line 288, define what “grass root level” means”

8.     In line 294, change “identification” to “identify”

9.     For Tables 2 and 3, use bullet points to mark each statement. It is difficult to distinguish the different points. Also, the points in the section on social in the tables are all underlined – this may be a mistake, but using the bullet points as suggested will help.

10.  In line 349, delete the word “been”

11.  Define “NOPs” the first time it is used. Also, “SAPs” is defined in the text, but much later than it is used in the tables. It would be helpful to define SAPs in the table.

12.  In line 472, change “consideration” to “considers”

13.  In line 523, change “caused” to “causes”

Author Response

Comments and Suggestions for Authors

Summary: The current manuscript by Adebiyi et al., aimed to investigate professionals’ impressions regarding policies on Fetal Alcohol Spectrum Disorders in South Africa. Overall, the authors found that there is concern and a clear need for new policies at various levels. The manuscript is a bit long because samples of many of the responses are provided. However, it is generally written well and is easy to follow. Below are a couple of suggestions to improve the quality and impact of this manuscript:

Query: 1. In line 69, what does “women in relation alcohol consumption” mean? Perhaps there is a missing word.

Response: Thanks for pointing this out.

Action: It has been corrected “The authors found that only two documents referred to FAS and another one to women and alcohol consumption during pregnancy” (line 68-69)

Query: 2.     In line 91, “and this factors” needs to be changed to “and these factors”

 Response: Agreed with the reviewer.

 Action: It has been corrected “these factors”

Query: 3.     It would useful to include the level education from the participants, such as highest degree earned, in addition to "profession" that is already provided, in the methods section

 Response: Agreed with the reviewer.

 Action: It has been added. “All the participants had tertiary qualifications as the lowest qualification.”’

Query: 4.     In line 169, “transcriptions” needs to be changed to “transcripts”

 Response: Agreed with the reviewer.

 Action: It has been changed. ‘’Verbatim transcripts of the participants’ responses were included to ensure credibility”.

Query: 5.     In line 241, the word “naysayers” is a strong word to describe those who feel that it would be resource intensive to put new policies in play. Perhaps using "individuals" would be more professional and less judgmental.

 Response: Agreed with the reviewer.

 Action: It has been changed. “individuals”

Query: 6.     In line 261, change “… regarding FASD, there are an ongoing awareness …” to “… regarding FASD that there is an ongoing …”

 Response: Thanks for pointing this out.

 Action: The sentence has been corrected. ''Regarding FASD, a participant from DSD reported that there are ongoing awareness and prevention programs on alcohol and drug abuse under the special program of the Department of Social Development.''

Query: 7.     In line 288, define what “grass root level” means”

 Response: Thanks for your observation.

 Action: It has been changed

Query: 8.     In line 294, change “identification” to “identify”

 Response: Thanks for your suggestion.

 Action: It has been changed.

Query: 9.     For Tables 2 and 3, use bullet points to mark each statement. It is difficult to distinguish the different points. Also, the points in the section on social in the tables are all underlined – this may be a mistake, but using the bullet points as suggested will help.

  Response: Thanks for pointing this out.

 Action: we have used another table format to differentiate the points because bullet points made the table longer

Query: 10.  In line 349, delete the word “been”

  Response: Thanks for the suggestion.

 Action: It has been deleted.

Query: 11.  Define “NOPs” the first time it is used. Also, “SAPs” is defined in the text, but much later than it is used in the tables. It would be helpful to define SAPs in the table.

  Response: Thanks for the suggestion.

 Action: It has been defined.

Query: 12.  In line 472, change “consideration” to “considers”

Response: Thanks for pointing this out.

Action: The sentence has been changed. 

''Furthermore, existing interventions that are specific to FASD are not addressed by the above policies.'' 

Reviewer 2 Report

Policy makers Perspectives Towards Developing a Guideline to Inform Policy on Fetal Alcohol Spectrum Disorder: A Qualitative Study

Abstract

Line 13 this would read better if the word exploratory and explored were not included in the same sentence.

The abstract could be edited to improve clarity.

Introduction

Line 29 Is alcohol the primary cause or a primary cause?

Line 30 FASD is both an umbrella term for the four categorical diagnoses in one diagnostic schema and a diagnostic entity for the adverse effects from prenatal alcohol exposure in another.

Line 35 Does lifetime consumption mean (ever drank)?

Line 42 What is the national average prevalence rate for FASD?

Line 44 could some of this change be due to differences in the study design?

Line 50 how about FASD is recognized as an important public….

Line 56 -58 Did you study people with alcohol or drug use disorders?

Line 63-64 This sentence is unclear to me.

Line 69 sentence needs rewritten

Line line 75 76 needs rewritten

Line 86-87 Is Dop a current problem? If not it might be useful to note this.

Line 91 this should be these.

Between line 97 and 105 It would be useful to share how many people you are referring to

Would you be interested in looking at the number of live births times the prevalence rate for FASD.

A second very useful estimate is the number of people with FASD birth -18 years of age. These estimates are often useful in clarifying the tasks ahead and the potential burden on service systems.

Line 103-104 I’m not sure we have evidence of this although it’s often stated.

Line 115 I’m not sure this fits. You did not describe interviews with relevant stakeholders for example mothers of children with FASD, women who drink during pregnancy, and importantly women who did not drink during pregnancy

The first 200 lines or so need careful proofreading.

In lines 203-204 was this the response from all people at the Department of Education, or just one.

Results

It seems to me that one of the key categories is the degree of consistency across reporters from individual departments. If you selected four people from education - were their responses consistent or did they differ widely.  This is often a problem within systems, some people are knowledgeable some are not - but often they all get to voice an opinion.

As an examplelines 203-204Is this an consistent finding or are the opinions widely discrepant.

Line 218 Cerebral

Line 247-248 This is a problem and not a solution.

Line 264-266 I wonder if this is different. FASD occurs after alcohol use during pregnancy. Many women we see are aware of the consequences of drinking during pregnancy but do not stop. The fact that they are aware is likely one of the reasons they deny alcohol use or underreport the amount of use.

Table 2 line 1 its hard to figure out if our concern should be confined to pregnant women with a “drinking problem” or any woman who drinks during pregnancy.

5 lines down

I’m not sure we have data to focus on women with alcohol abuse but may want to concentrate on women who drink during pregnancy.

Line 349 around the world only 1-5% of FASD has been identified. So it seems unlikely that over-identification is an issue.

Line 478-481 This is correct but I’m not sure most people understand what this means and why its important.

Firstly it’s important because South Africa is currently spending huge amounts of money on this population.

Secondly, it’s important because since they are undiagnosed, agencies are treating the wrong problem.

Thirdly, often they are treating and trying to prevent the outcomes from having FASD rather than FASD itself.

Fourthly, much of the money being spent is currently in an agency’s budget and is wasted. This means we don’t have money to change what we are doing. This is the importance of diagnosis. A correct diagnosis can help track cases, service utilization, and costs. Correct diagnosis also helps provide the rational for workforce development which is essential for a condition like FASD.

Line 504-506 This could be correct but also what about treating something else, not using evidence based interventions, corruption (same agency always gets the money even if they offer very low quality services).

Line 535 SA has a huge number of children and adolescents with FASD. This model may work for a low prevalence disorder but is unlikely to be able to reach the majority of children with FASD.

Limitations

This section needs to be expanded. This type of study emphasizes opinion from a small group of people. It’s unknown if they have expertise on FASD. Their opinions may not be representative of the agency they represent. This may be a very biased sample or may represent the best expertise available.

This study did not include any women who drank during pregnancy, any woman with a child who has FASD or women who went through pregnancy and did not drink (very important group). It does not include any subjects with FASD or women with prenatal alcohol use who quit drinking upon hearing they were pregnant. No one commented on the role of men in changing behavior of women or in caring for women with alcohol use disorders or children with FASD.

This is extremely important work. Thank you for doing it.

I think it would be useful for you to consider adding a FASD fact sheet to your report for policy makers.

Annual cases of FASD in South Africa (prevalence X annual births)

(annual cases X 18)

FASD Birth -18

Maternal mortality events

Stillbirths

Neonatal deaths

Infant deaths

Childhood mortality

Recurrent births

Annual cost of care per case

Lifetime costs of care per case

FASD: annual costs for South Africa.

I hope you will consider these comments in revising your paper. These comments are offered to improve your paper.

Author Response

Comments and Suggestions for Authors

Policymakers Perspectives Towards Developing a Guideline to Inform Policy on Fetal Alcohol Spectrum Disorder: A Qualitative Study

Abstract

Query: Line 13 this would read better if the word exploratory and explored were not included in the same sentence.

Response: Thank you for your suggestion.

Action: We have rephrased the sentence to avoid the double use of explore.

“Using a qualitative study design, we explored the perspectives of policymakers on guidelines/policies for FASD, current practices, and interventions and what practices and interventions that could be included in a policy for FASD.”

Query: The abstract could be edited to improve clarity.

Response: Agreed with the reviewer

Action: The abstract has been edited for clarity (line 11-24)

Introduction

Query: Line 29 Is alcohol the primary cause or a primary cause?

Response: Thanks for your observation

Action: We have adjusted the sentence by writing: “the primary cause” (line 29)

Query: Line 30 FASD is both an umbrella term for the four categorical diagnoses in one diagnostic schema and a diagnostic entity for the adverse effects from prenatal alcohol exposure in another.

Response: Agreed with the reviewer

Action: It has been added (line 30-31)

Line 35 Does lifetime consumption mean (ever drank)?

Response: Thanks for your observation. Yes

Action: No action was taken (line 37)

Query: Line 42 What is the national average prevalence rate for FASD?

Response: Thanks for your question.

Action: The national average will be 290 + 29 divided by 2 = 159.5. However, it has always been reported in a range (line 39)

Query: Line 44 could some of this change be due to differences in the study design?

Response: Thanks for your observation. No

Action: Study designs were similar.

Query: Line 50 how about FASD is recognized as an important public….

Response: Thanks for the suggestion

Action: It has been added “FASD is recognized as an important public health problem in South Africa”

Query: Line 56 -58 Did you study people with alcohol or drug use disorders?

Response: No

Action: No action was taken

Query: Line 63-64 This sentence is unclear to me.

Response: Thanks for pointing this out

Action:  No action was taken “Jacobs et al. [14] acknowledged that efforts were being made to develop relevant policies to address the use of alcoholic beverages by pregnant women in South Africa, but the authors stressed the need to address the gaps related to the conspicuous absence of issues around FASD.” (line 62-65)

Line 69 sentence needs rewritten

Response: Thank you for the suggestion

Action: The sentence has been rewritten: “The authors found that only two documents referred to FAS and another one to women and alcohol consumption during pregnancy.” (line 68-69)

Query: Line 75 76 needs rewritten

Response: Thank you for the suggestion

Action:  The sentence has been rewritten: “In light of the absence of a coordinated effort targeting FASD, we identified the need to develop a guideline to inform a multi-sectoral and interdepartmental policy to address FASD. In this article, we report on a step towards developing a guideline to inform a coordinated approach in the prevention and management of FASD” (line 74-77)

Query: Line 86-87 Is Dop a current problem? If not it might be useful to note this.

Response: Thanks for pointing this out

Action: We added this “Although the system has been abolished, the lingering effects remain.” (line 87-88)

Query: Line 91 this should be these.

Response: Agreed with the reviewer

Action: It has been changed (line 91)

Query: Between line 97 and 105 It would be useful to share how many people you are referring to

Response: Thanks for pointing this out

Action: It has been corrected “According to a study conducted in the Cape Metropole, Western Cape, out of 684 pregnant women sampled, 36.9% confirmed that they had consumed alcohol during their current pregnancy or in the three months before they knew they were pregnant [29].”

“In another study conducted in Cape Town with 110 samples (parents or caretakers of 110 children aged 4 - 12 years), the reported alcohol use six months before pregnancy and during pregnancy was 38% and 15%, respectively [30].’’

Query: Would you be interested in looking at the number of live births times the prevalence rate for FASD.

Response: Thanks for suggesting this. No, maybe in another study

Action: It is not part of the aim of our study

Query: A second very useful estimate is the number of people with FASD birth -18 years of age. These estimates are often useful in clarifying the tasks ahead and the potential burden on service systems.

Response: Thanks for suggesting this. No, maybe in another study

Action: This is not readily available in the literature. Also, it is not the aim of our study

Query: Line 103-104 I’m not sure we have evidence of this although it’s often stated.

Response: Thanks for pointing this out. Yes

Action: The authors made reference to it. (line 103-104)

Query: Line 115 I’m not sure this fits. You did not describe interviews with relevant stakeholders for example mothers of children with FASD, women who drink during pregnancy, and importantly women who did not drink during pregnancy

 Response: Thanks for pointing this out. In this study, policymaker are the relevant stakeholders. I didn’t mention anyone in the sentence because I was discussing what it is qualitative research

Action: This study is part of a larger study. The aim of this part is to explore the perspectives of policymakers. Therefore, mothers of children with FASD, women who drink during pregnancy and women who did not drink during pregnancy were not interviewed. There was another study, which focuses on service providers, some of which are foster parents of individuals with FASD. (Adebiyi et al., 2018) (line 115)

Query: The first 200 lines or so need careful proofreading.

 Response: Thanks for pointing this out.

Action: It has been proofread by an editor. (line 1-200)

Query: In lines 203-204 was this the response from all people at the Department of Education, or just one.

Response: One

Action: However, each participant represents its department. Therefore, when any one of them is speaking, he/she speaks about what it is obtainable in his/her department.

Results

Query: It seems to me that one of the key categories is the degree of consistency across reporters from individual departments. If you selected four people from education - were their responses consistent or did they differ widely.  This is often a problem within systems, some people are knowledgeable some are not - but often they all get to voice an opinion

Response: Thanks for pointing this out

Action: Their response different on some issues, especially issues that require a personal opinion.

Query: As an example lines 203-204 Is this an consistent finding or are the opinions widely discrepant.

Response: Agreed with the reviewer

Action: It is an example of consistency in opinion.                                                                                                    

Query: Line 218 Cerebral

Response: Agreed with the reviewer

Action: It has been corrected

Query: Line 247-248 This is a problem and not a solution.

Response: Agreed with the reviewer

Action: It is part of the explanation gave for why a separate policy may not be necessary.

Query: Line 264-266 I wonder if this is different. FASD occurs after alcohol use during pregnancy. Many women we see are aware of the consequences of drinking during pregnancy but do not stop. The fact that they are aware is likely one of the reasons they deny alcohol use or underreport the amount of use.

 Response: Agreed with the reviewer

Action:  The participant is explaining the services that they are providing as a department

Query: Table 2 line 1 its hard to figure out if our concern should be confined to pregnant women with a “drinking problem” or any woman who drinks during pregnancy.

Response: Agreed with the reviewer

Action:  The contents of the table are derived from what the participants are saying. The table is a summary of their responses.

Query: 5 lines down

I’m not sure we have data to focus on women with alcohol abuse but may want to concentrate on women who drink during pregnancy.

Response: Agreed with the reviewer

Action: Noted

Query: Line 349 around the world only 1-5% of FASD has been identified. So it seems unlikely that over-identification is an issue.

Response: Agreed with the reviewer

Action:  There was an issue raised by one of the participants “not to put the diagnosis in the hands of the teachers” (line 357). The statement was made in reference to that. It does not connote over-identification. The statement emphasized on proper diagnosis.

Query: Line 478-481 This is correct but I’m not sure most people understand what this means and why its important.

Response: Thanks for pointing this out

Action: Many people will understand if they read the article cited.

Query: Firstly it’s important because South Africa is currently spending huge amounts of money on this population.

Response: Thanks for pointing this out.

Action: Many people will understand if they read the article cited.

Query: Secondly, it’s important because since they are undiagnosed, agencies are treating the wrong problem.

Response: Thanks for pointing this out.

Action: Many people will understand if they read the article cited.

Query: Thirdly, often they are treating and trying to prevent the outcomes from having FASD rather than FASD itself.

Response: Thanks for pointing this out.

Action: Many people will understand if they read the article cited.

Query: Fourthly, much of the money being spent is currently in an agency’s budget and is wasted. This means we don’t have money to change what we are doing. This is the importance of diagnosis. A correct diagnosis can help track cases, service utilization, and costs. Correct diagnosis also helps provide the rational for workforce development which is essential for a condition like FASD.

Response: Thanks for pointing this out.

Action: No action was taken.

Query: Line 504-506 This could be correct but also what about treating something else, not using evidence based interventions, corruption (same agency always gets the money even if they offer very low quality services).

Response: Thanks for pointing this out.

Action: No action was taken.

Query: Line 535 SA has a huge number of children and adolescents with FASD. This model may work for a low prevalence disorder but is unlikely to be able to reach the majority of children with FASD.

Response: Thanks for your observation

Action: This is part of a larger study toward developing the guideline. Other studies that made up the larger study will include other aspects.

Limitations

Query: This section needs to be expanded. This type of study emphasizes opinion from a small group of people. It’s unknown if they have expertise on FASD. Their opinions may not be representative of the agency they represent. This may be a very biased sample or may represent the best expertise available.

This study did not include any women who drank during pregnancy, any woman with a child who has FASD or women who went through pregnancy and did not drink (very important group). It does not include any subjects with FASD or women with prenatal alcohol use who quit drinking upon hearing they were pregnant. No one commented on the role of men in changing behavior of women or in caring for women with alcohol use disorders or children with FASD.

 Response: Thanks for the observation

Action: The limitation has been expended.

“In this study, the researcher interviewed policymakers from the three major Departments (Education, Health, and Social Development) that are responsible for the implementation of policies and the provision of services to individuals with FASD. A limitation of the study that was expected to inform policy on FASD nationally was that it was only conducted in one of nine provinces in South Africa. We also did not interview policymakers from the Department of Labor and the Department of Justice, who may have had additional information to improve the guideline. Furthermore, we used purposeful sampling in the study, which may have prompted biased opinions. Moreover, since the focus of this study was on policymakers, we did not include women who drank during pregnancy, women with children who have FASD, women who went through pregnancy and did not drink, individuals with FASD or women with prenatal alcohol use who quit drinking upon hearing they were pregnant”. (line 584-593)